# A gas-plastic elastomer that quickly self-heals damage with the aid of $CO_2$ gas

Yohei Miwa [1], Kenjiro Taira[1], Junosuke Kurachi[1], Taro Udagawa [1] & Shoichi Kutsumizu[1]

Self-healing materials are highly desirable because they allow products to maintain their performance. Typical stimuli used for self-healing are heat and light, despite being unsuitable for materials used in certain products as heat can damage other components, and light cannot reach materials located within a product or device. To address these issues, here we show a gas-plastic elastomer with an ionically crosslinked silicone network that quickly self-heals damage in the presence of $CO_2$ gas at normal pressures and room temperature. While a strong elastomer generally exhibits slow self-healing properties, $CO_2$ effectively softened ionic crosslinks in the proposed elastomer, and network rearrangement was promoted. Consequently, self-healing was dramatically accelerated by ~10-fold. Moreover, self-healing was achieved even at $-20\,°C$ in the presence of $CO_2$ and the original mechanical strength was quickly re-established during the exchange of $CO_2$ with air.

[1] Department of Chemistry and Biomolecular Science, Faculty of Engineering, Gifu University, Yanagido, Gifu 501-1193, Japan. Correspondence and requests for materials should be addressed to Y.M. (email: y_miwa@gifu-u.ac.jp)

Nature produces a multitude of elegant dynamic networks that perform specific functions, such as self-healing, excellent fatigue and fracture resistance, and ultra-stretchability. For example, the dynamic crosslink properties of actin filaments in cytoskeletal networks demonstrate certain cell activities, such as adaption and migration of cells[1,2]. Another example is spider silk, where, upon external loading, densely assembled hydrogen bonds in the silk repeatedly deform and reform to effectively dissipate energy, avoiding the concentration of local stresses that trigger fractures[3,4]. The concept of dynamic networks has the potential to play an important role in improving safety, energy efficiency, mechanical toughness, and the lifetime of human-made products[5]. In particular, self-healing materials that perform at ambient temperatures are exceptionally useful for specific types of futuristic high-performance products, such as artificial skin, soft robots, wearable sensors and PCs, and smart actuators, which will have the capability to maintain their performance for humans in daily life[6–8].

Dynamic networks autonomously self-heal at room temperature in cases where flexible polymers with low glass transition temperatures ($T_g$) are weakly crosslinked by dynamic bonds, such as hydrogen bonding[9–15], metal–ligand coordination[16–19], and ionic bonding[20–23], even without the application of healing agents or the input of external energy. However, these materials often present a trade-off between strength and their autonomous self-healing rate, i.e., self-healing materials that respond quickly are generally viscoelastic and exhibit low tensile strength, often even far below 1 MPa. In contrast, thermally or optically reversible bonds and interactions display rapid self-healing characteristics while maintaining a high degree of strength[24–35]. However, heat and light, which are the most commonly used stimuli for self-healing, are unsuitable for materials used in certain products because heat can potentially damage other components, and light is not applicable if those materials are located inside the product.

To address these issues, a dynamic network that quickly self-heals damage at ambient temperature with the aid of $CO_2$ gas is presented. The ionically crosslinked poly(dimethyl siloxane) (PDMS) elastomer described in this paper exhibited high strength (~3.5 MPa) in air, though the rearrangement of the network was dramatically accelerated by $CO_2$ gas, which effectively softens the material's crosslinking sites, such as the ionic aggregates (Fig. 1a, c). This results in rapid self-healing in $CO_2$ gas atmosphere at normal pressures (~0.1 MPa) (Fig. 1d). Notably, the original mechanical strength was quickly re-established in the exchange of $CO_2$ with air (Fig. 2b); i.e., the proposed material is the gas-plastic elastomer, which makes it particularly useful in the development of self-healing materials because the gas is able to permeate components inside the product, and the exposure of most products to $CO_2$ gas would not result in damage to other components. Our gas-plastic elastomer is based on this conceptual design.

## Results

**Material design**. We synthesized an oligomeric PDMS, which has COOH groups at the ends of the chain and at middle positions placed randomly along the backbone; the COOH groups are partially neutralized with sodium (Fig. 1a and Supplementary Fig. 1). PDMS was chosen as a backbone because of its flexibility, biocompatibility, and high $CO_2$ permeability[36]. In this case, low-molecular-weight PDMS is suitable because self-healing of the dynamic network is accomplished via chain diffusion of constituent polymers. The number average molecular weight and

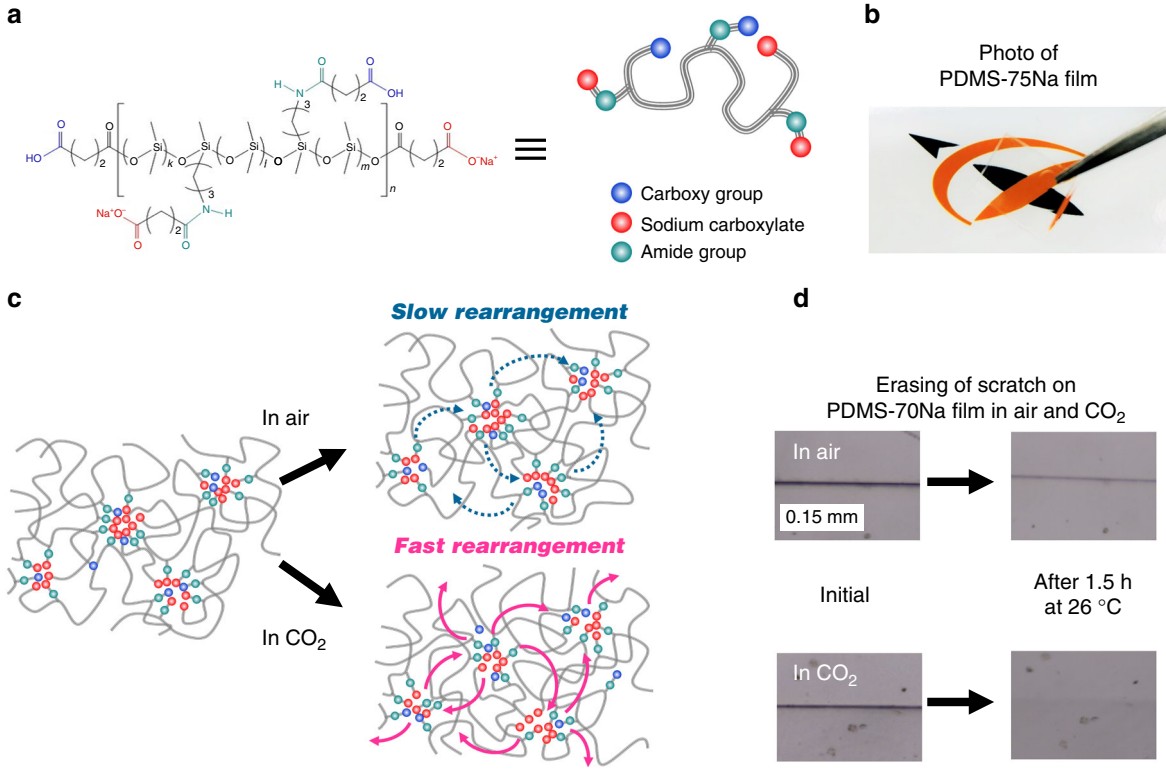

**Fig. 1** Molecular structure and network rearrangement chemistry of PDMS-xNa. **a** Chemical structure and schematic illustration of PDMS-xNa. **b** Photograph of a transparent and colorless PDMS-75Na film. **c** Schematic illustration for slow and fast rearrangement of ionic crosslinks in air and $CO_2$, respectively. Plasticization of ionic aggregate by $CO_2$ gas provides rapid network rearrangement. **d** Optical microscopic images of a razor scratch on a PDMS-70Na film surface healed at 26 °C for 1.5 h in air and in $CO_2$

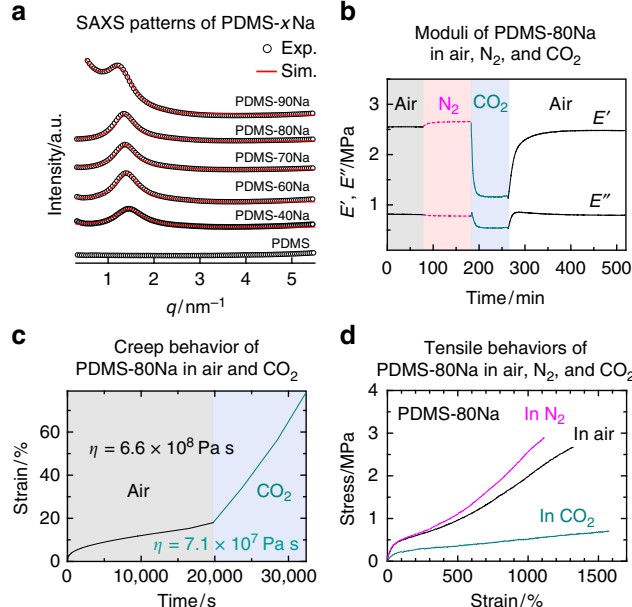

**Fig. 2** Effect of $CO_2$ on mechanical properties of PDMS-80Na.
**a** Experimental and simulated SAXS patterns for PDMS-$x$Na. **b** Effect of each gas on the storage modulus ($E'$) and loss modulus ($E''$) for PDMS-80Na measured at 1 Hz and 30 °C. **c** Creep test for PDMS-80Na measured at a stress of 4 kPa at 30 °C. Air flow is changed to $CO_2$ gas flow. **d** Tensile tests for PDMS-80Na measured at 27 ± 1 °C in air, in $N_2$, and in $CO_2$. The stretching speed is 100 mm min$^{-1}$

the molecular weight distribution of the PDMS are 7300 and 1.56, respectively. However, a decrease in the molecular weight tends to result in the display of weak and viscoelastic properties in the particular material. Therefore, the ends of the chain were capped with an ionic group to restrict their mobility and thereby reduce this tendency. The total COOH concentration was determined to be 3.8 mol% by titration, and 40–90% of the COOH groups were neutralized with sodium hydrate. The unneutralized and neutralized PDMSs are denoted as PDMS–COOH and PDMS-$x$Na, respectively, where $x$ indicates the percentage of neutralization. These materials were characterized by gel permeation chromatography, $^1$H-NMR (Supplementary Fig. 2), and Fourier-transform infrared spectroscopy (FT-IR) (Supplementary Fig. 3). The PDMS-$x$Na film is completely colorless and highly transparent (Fig. 1b).

The size of the ionic aggregates acting as physical crosslinks in PDMS-$x$Na was determined by the simulation of small-angle X-ray scattering (SAXS) pattern based on the Yarusso–Cooper model[37] (Fig. 2a). In this model, spherical-shaped ionic aggregates with a radius of $R_1$ are randomly dispersed subject to the closest approach limitation, $2R_{CA}$. $R_1$ (1.2 nm), $R_{CA}$ (2.0 nm), and the number density (24 (10 nm)$^{-3}$) of the ionic aggregates in PDMS-$x$Na are virtually independent of the level of neutralization, except for PDMS-90Na (Supplementary Fig. 4). This suggests that most of the neutralized and unneutralized COOH groups are involved in ionic aggregates irrespective of the level of neutralization. When the $T_g$ of backbone polymers is far below room temperature, the neutralized and unneutralized COOH groups dynamically hop between ionic aggregates at room temperature;[23,38,39] in other words, the network is considered to be dynamic. The hopping rate is determined based on the balance between the backbone polymer flexibility (mobility) and the attractive force between the crosslinking sites. Fast rearrangement of ionic crosslinks results in rapid self-healing, although it is also the cause of weak and viscoelastic material properties. PDMS

($T_g \approx -120$ °C) is significantly flexible polymer (Supplementary Fig. 5). Therefore, not only ionic groups, but also hydrogen-bonding amide groups were additionally introduced to enhance the attractive force between crosslinking sites in our designed PDMS-$x$Na (Fig. 1a). The interaction energy of the hydrogen bonding between the amide groups was determined to be −78.7 kJ mol$^{-1}$ based on density functional theory calculations, where the energy between sodium carboxylates was found to be −190 kJ mol$^{-1}$ (Supplementary Fig. 6).

**Mechanical properties.** In contrast to other network systems crosslinked by weak and dynamic bonds, such as hydrogen bonding[9–15] and metal–ligand coordination[16–19], the rearrangement rate of ionic crosslinks is easily tunable by altering the neutralization level. The relaxation temperature assigned to the network rearrangement in the PDMS-$x$Na decreased as the neutralization level decreased (Supplementary Fig. 7). Namely, the rearrangement rate of ionic crosslinks in PDMS-$x$Na at room temperature increased as the neutralization level decreased because unneutralized carboxy groups act as plasticizers in the ionic aggregates[39].

The rearrangement of ionic crosslinks simultaneously performs several important mechanical functions. It enhances the fracture resistance due to the detachment of stressed chains, and therefore toughens the elastomer, while it also enhances the fatigue resistance and autonomic self-healing properties at room temperature without the input of energy (e.g., heat or light). Due to the rearrangement of the ionic crosslinks, the strength and stretchability of PDMS-$x$Na severely depend on the neutralization level and the stretching speed (Supplementary Fig. 8). For example, when the neutralization level is high (80%), a high degree of fracture stress (~3.5 MPa) is displayed for stretching at 300 mm min$^{-1}$. On the other hand, very high stretchability (~12,000%) is achieved for a relatively low neutralization level (40%) at 10 mm min$^{-1}$, while a very high neutralization rate (90%) produces a brittle film. For oligomeric PDMS with very high neutralization level, intramolecular aggregation of ionic groups is probably enhanced and crosslinking polymer chains between ionic aggregates are decreased, thereby yielding brittle PDMS-90Na.

**Gas-plastic properties.** The PDMS-$x$Na presented in this study actually exhibited gas-plastic properties. For example, in $CO_2$ gas atmosphere at normal pressures, the storage modulus ($E'$) of PDMS-80Na dropped steeply to less than half of that in air, while it slightly increased in $N_2$ gas (Fig. 2b), which indicated that the $CO_2$ gas softened the material. On the other hand, the $E'$ quickly recovered as a result of the exchange of $CO_2$ with air. Moreover, when a stress of $4 \times 10^3$ Pa was applied to PDMS-80Na, the strain rate increased from 2.2% per hour in air to 20% per hour in $CO_2$ (Fig. 2c). At the same time, the fracture stress of PDMS-80Na decreased significantly from 2.7 MPa in air to 0.7 MPa in $CO_2$ (Fig. 2d). Some polymers, including PDMS, exhibit $CO_2$ gas plasticization[40–42]. However, our PDMS-$x$Na elastomer exhibits more distinct $CO_2$ gas plasticization. In fact, the moduli of commercially available chemically crosslinked PDMS elastomer displayed little change from the application of $CO_2$ gas (Supplementary Fig. 9). Our results clearly demonstrated two important characteristics: (1) $CO_2$ gas affects ionic domain and (2) high $CO_2$ permeability of PDMS produces a rapid self-healing response. Notably, moisture also plasticizes PDMS-80Na; however, $CO_2$ gas plasticizes PDMS-80Na more efficiently (Supplementary Fig. 10).

The FT-IR bands at 1586 cm$^{-1}$ and 1716 cm$^{-1}$ assigned to the stretching vibrations of sodium carboxylate and carboxy groups, respectively, are affected by $CO_2$ exposure (Fig. 3a). This demonstrates that the $CO_2$ molecules having relatively high polarizability penetrate into the ionic aggregates, leading to the

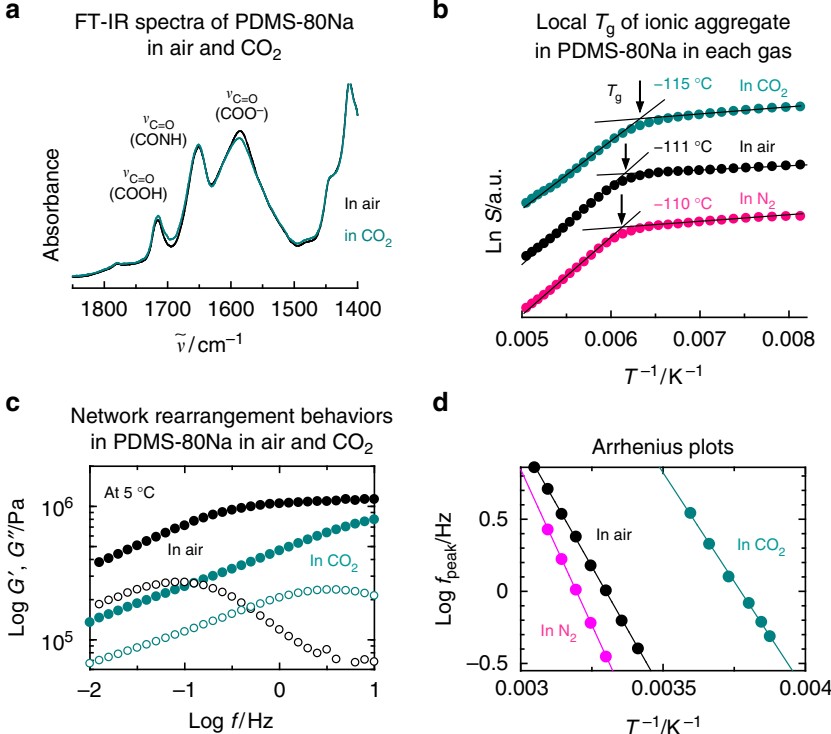

**Fig. 3** Effect of $CO_2$ on ionic aggregate and network rearrangement. **a** FT-IR spectra of PDMS-80Na measured in air and in $CO_2$. **b** Temperature dependence of the saturation factor ($S$) for PDMS-80Na in air, $N_2$, and $CO_2$. The inflection temperature represents the local $T_g$ of ionic aggregate. **c** Frequency sweep of storage modulus ($G'$, solid symbols) and loss modulus ($G''$, open symbols) for PDMS-80Na measured at 5 °C in air and in $CO_2$. The relaxation is assigned to rearrangement of ionic crosslinks. **d** Arrhenius plots of peak frequency of $G''$ for PDMS-80Na in air, in $N_2$, and in $CO_2$

plasticization of the ionic aggregates. The plasticization of ionic aggregates by $CO_2$ gas was directly confirmed via an electron spin resonance (ESR) technique reported by Miwa et al.[43]. When using this method, the local $T_g$ of the ionic aggregates in the PDMS-xNa is selectively measured by ESR using a spin probe localized in the ionic aggregates, and here, 4-carboxy-TEMPO was used (Supplementary Fig. 11). The location of the spin probe in the ionic aggregates was confirmed from one of the magnetic parameters of nitroxide, $A_{zz}$, which increased with increasing local polarity. The large $A_{zz}$ value of 34.7 Gauss was evidence of 4-carboxy-TEMPO located in the ionic aggregates of PDMS-80Na[43]. In $CO_2$, the local $T_g$ of ionic aggregates in PDMS-80Na decreased by 4 °C and 5 °C compared with that in air and in $N_2$, respectively (Fig. 3b). The plasticization of ionic aggregates resulted in an increase in the network rearrangement rate. In fact, the relaxation assigned to the network rearrangement in PDMS-80Na was distinctly accelerated in $CO_2$ gas atmosphere, while the relaxation rate slowed slightly in $N_2$, compared with that in air (Fig. 3c, d; Supplementary Fig. 12). The relaxation time, $\tau$ ($= f_{peak}^{-1}$), of the network rearrangement at 25 °C was determined to 1.6 s, 4.9 s, and 0.055 s in air, in $N_2$, and in $CO_2$, respectively (Fig. 3d). The network rearrangement accelerated by approximately 30 times in the presence of $CO_2$ gas compared with that in air.

**Self-healing properties**. PDMS-xNa exhibited autonomic self-healing at room temperature. For example, when a cherry blossom-shaped PDMS-75Na film was cut, placed in contact, and stored at room temperature, the cut pieces completely connected (Fig. 4a). The PDMS-xNa self-heals irrespective of the neutralization level (Supplementary Fig. 13). Under $CO_2$ exposure, the self-healing quickly undergoes (Fig. 4b). The self-healing behavior of PDMS-80Na was accelerated by ~10 times in $CO_2$ compared with that in air because of the accelerated network rearrangement (Fig. 4c), and

healing was achieved even at −20 °C with a healing efficiency of ~50% after 1 week (Fig. 4d). At −10 °C, healing efficiencies of ~50% and ~90% were achieved after 3 days and 1 week, respectively. Unlike other self-healing materials that chemically consume $CO_2$ gas, such as concrete[44] and a hydrogel containing chloroplast[45], our gas-plastic elastomer physically utilizes $CO_2$ gas as a plasticizer. Therefore, this elastomer permanently demonstrates the rapid self-healing induced by $CO_2$ exposure.

## Discussion
In this study, we developed a gas-plastic silicone elastomer that exhibits high mechanical strength and rapid self-healing with the aid of $CO_2$ gas. The key aspect of the gas-plastic capability is the softening of the ionic aggregates that act as crosslinking sites in the elastomer owing to the effects of the $CO_2$ gas. Importantly, the gas-plastic technique should be widely applicable for other ion-containing polymeric materials. This would be practical for not only self-healing but also for saving energy because the gas-plastic capability allows polymers to mold at much lower temperatures in the presence of $CO_2$. Further studies are currently underway in our laboratory and will be reported in due course.

## Methods
**Synthesis of PDMS-xNa**. The PDMS-xNa elastomer was prepared via carboxylation of PDMS, and then the carboxy groups were neutralized to some degree using sodium hydroxide (Supplementary Fig. 1). The PDMS containing amino groups at middle positions placed randomly along the backbone (PDMS–NH$_2$) was synthesized via hydrolysis and condensation polymerization of diethoxydimethylsilane and 3-aminopropyldiethoxymethylsilane. The actual amino group's concentration in the PDMS–NH$_2$ calculated from $^1$H-NMR spectrum was ~2.7 mol% (Supplementary Fig. 2a). The number average molecular weight and molecular weight distribution of the PDMS–NH$_2$ determined using polystyrene standards were 7300 and 1.56, respectively. The PDMS–NH$_2$ was reacted with succinic anhydride in dry chloroform. The actual concentration of the COOH group in PDMS–COOH was determined via neutralization titration with NaOH/

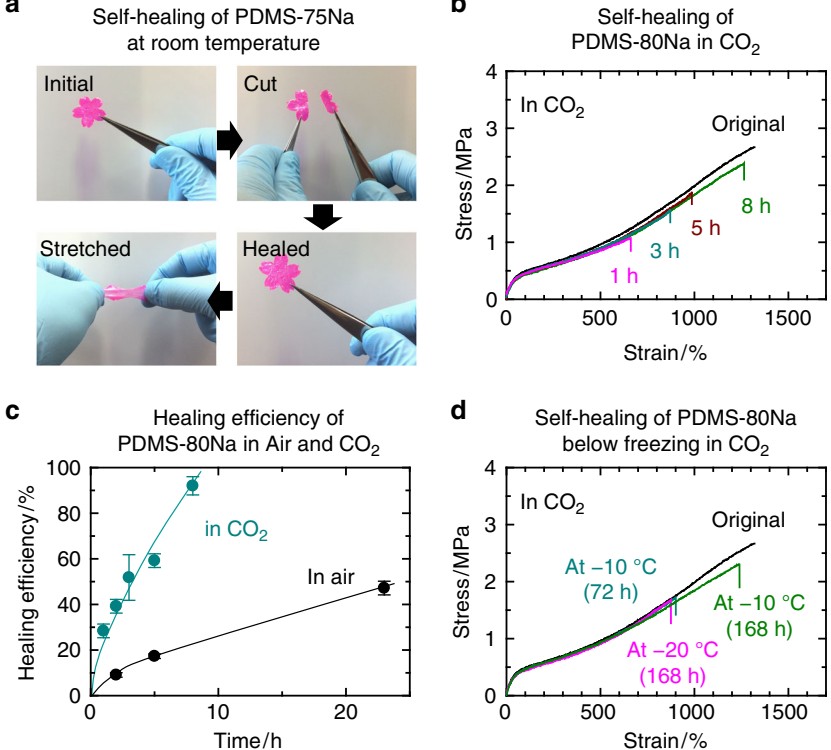

**Fig. 4** Effect of $CO_2$ on self-healing of PDMS-$x$Na. **a** Photograph of self-healing behavior of cherry blossom-shaped PDMS-75Na film at 26 °C. The film was painted pink. **b** Stress−strain curves of PDMS-80Na healed at 26 °C in $CO_2$. **c** Time variation of healing efficiency of PDMS-80Na stored at 26 °C in air and in $CO_2$. The error bars indicate standard deviations. **d** Stress−strain curves of PDMS-80Na healed at −10 °C and −20 °C in $CO_2$

methanol solution, and the concentration of the COOH group was 3.8 mol%, which is higher than that of the $NH_2$ group in the PDMS−$NH_2$; this is because succinic anhydride was added to the hydroxy chain ends. An appropriate amount of NaOH/methanol solution was slowly added to a PDMS−COOH/THF solution (7 wt%) under vigorous stirring. The mixture was then poured into a Teflon petri dish and dried at 35 °C to form a cast film. The cast film was further dried at 35 °C for more than 1 day in a vacuum. Upon neutralization, the intensity of the COOH carbonyl stretching band at 1716 cm$^{-1}$ decreases, and a band indicating sodium carboxylate's stretching vibration is generated at 1586 cm$^{-1}$ in the Fourier-transform infrared spectra (Supplementary Fig. 3).

Synchrotron SAXS measurements were performed using the BL-6A beamline at the Photon Factory of the High Energy Accelerator Research Organization (KEK) in Tsukuba, Japan. The X-ray wavelength ($\lambda$) was 0.15 nm. Stearic acid and silver behenate were used as SAXS detector calibration standards. The intensities were radially integrated, averaged, and redistributed to convert the pixel number to the corresponding scattering vector $q$ ($q = (4\pi/\lambda)\sin\theta$) and produce a circularly averaged pattern.

Dynamic mechanical measurements were performed in tensile mode on a TA Instruments DMA Q800. Isothermal measurements were performed in each gas atmosphere at 1 Hz and 30 °C. The rectangular specimen dimensions were $10 \times 4.5 \times 0.4$ mm, and a strain of 0.5% was applied. Each gas (dry air, $N_2$, and $CO_2$) was flown into the sample chamber at a rate of 1.6 L min$^{-1}$ during the measurements. Creep measurements were performed with a stress of 4 kPa.

The rheological properties were investigated in oscillatory shear on a parallel-plate rheometer (AR-G2, TA instruments) with 8-mm-diameter plates. The sample thickness was ~0.4 mm, and a strain of 0.5% was applied. A temperature sweep test was conducted at 1 Hz in the −150 °C to 150 °C range at a heating rate of 3 °C min$^{-1}$. Frequency sweep tests were performed in each gas atmosphere (dry air, $N_2$, and $CO_2$) within a dynamic range of 0.01–50 Hz. Prior to the frequency sweep measurements, the dried PDMS-80Na film was exposed to an atmosphere of the relevant gas for 2 h. The moisture-absorbed PDMS-80Na was measured in air. Prior to the measurement, a PDMS-80Na film was stored for 5 days in a glass desiccator maintained at 75% humidity and 28 °C. The weight of the film increased by 2% because of the moisture absorption.

The sample films' tensile stress−strain curves were collected using the AND Force Tester MCT-2150 at 27 ± 1 °C under various gases (dry air, $N_2$, and $CO_2$). Dumbbell-shaped tensile bars, with dimensions of $25 \times 2.0 \times 0.4$ mm, were cut from the cast films. The initial gauge length was typically set to 11 mm. Each measurement was performed at least thrice. The tensile stress ($\sigma$) was calculated as $\sigma = F/S_0$, where $F$ is the loading force, and $S_0$ is the initial cross-sectional area of the sample film. The strain ($\varepsilon$) under elongation was defined as the marker distance ($l$) relative to the initial marker distance ($l_0$) of the specimen, i.e., $\varepsilon = (l − l_0)/l_0 \times 100\%$. The increasing marker distance was monitored by a video camera.

**Self-healing tests**. The self-healing of scratches on the PDMS-70Na film surface was monitored by an optical microscope (Olympus BX53P). The film was scratched using a razor, and the depth of the scratch was controlled to ~0.15 mm using a spacer. Scratched PDMS-70Na films were separately mounted on slide glass and stored at room temperature (26 °C) in either dry air or dry $CO_2$ atmosphere. The strength recovery of the self-healed films was measured via tensile testing at 27 ± 1 °C. A PDMS-80Na film was scored, leaving a thickness of 12.5 μm, using a razor and a spacer to avoid completely cutting the film into two separate pieces. The cut faces were then placed in contact. The PDMS-80Na films were stored at room temperature (26 °C) in either dry air or $CO_2$ for different periods. The healed PDMS-80Na films were then stretched at 100 mm min$^{-1}$ in air. The self-healing efficiency was calculated as the ratio between the tension energies required to break the original and self-healed materials. The tension energies were measured as the area below the stress−strain curve.

## Data availability

The authors declare that the data supporting the findings of this study are available within the article and its Supplementary Information files or are available from the authors upon reasonable request.

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

## Acknowledgements

Beam time at PF-KEK provided by Programs 2016G627 and 2017G562 is acknowledged herein. Portion of these DFT calculations was performed at Research Center for Computational Science (RCCS), Okazaki. This research was financially supported by the Japan Society for the Promotion of Science (Grant-in-Aid for Young Scientists (B), 16K17958 and Scientific Research (C), 16K05748); The Mazda Foundation (17kk-079); The Ogasawara Foundation for the Promotion of Science & Engineering; and The Koshiyama Foundation.

## Author contributions

Y.M. and S.K. planned and directed the project; Y.M., K.T., J.K., and T.U. conducted the experiments. Y.M. and K.T. analyzed the data; Y.M., S.K., and T.U. wrote the paper.

## Additional information

**Competing interests:** The authors declare no competing interests.

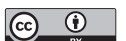

