## [Peer Review File · Nature Communications]

Reviewers' comments:

Reviewer #1 (Remarks to the Author):

This manuscript reports on an elastomer that plasticized and self-healed by exposure to CO₂ gas. To my knowledge, this is the first report on a polymer that plasticized by a gas compound. Even though self-healing induced by pure gas compounds including CO₂ may not have high potential for practical use because such stimuli is not ubiquitous, processability at low temperature is appealing in terms of energy-saving process. More importantly, the "gas-plastic" behavior is scientifically valued and of high impact due to its novelty. I believe this manuscript can be appropriate for publication in Nature Communications after addressing the comments listed below:

1. The "Gas-plastic" property of the elastomer including self-healing behavior is clearly demonstrated with many experimental data. However, unfortunately, the authors pay less attention to the mechanism of the Gas-plastic behavior. Why is this elastomer softened by CO₂ gas? The schematic illustration in Fig. 1c only show the experimental results that CO₂ enhance the rearrangement rate of ionic crosslinks. The reason why this happens should be explained. Also, why does the rearrangement rate of ionic crosslinks change depending on the neutralization level?
2. Why is PDMS-90Na so brittle compared with the other elastomers?
3. Healing test results are only given for PDMS-75Na and PDMS-80Na. Healing ability of the other samples should be also discussed.
4. Exact pressure of the experimental conditions including healing conditions should be provided. "Normal pressure" (line 2 on page 6) is unclear.
5. The chemical structures in Fig. 1a, S1, and S2 seems to be incorrect. For example, the structural formula given in Fig. S1 indicates that every polymer chain of PDMS-NH₂ has only one aminopropylmethylsilane in the middle of the chain. However, I guess from the preparation method that PDMS-NH₂ is a simple random copolymer of dimethylsilane and aminopropylmethylsilane containing 2.7 mol% of the latter monomer on average.

Reviewer #2 (Remarks to the Author):

I have read the manuscript "a gas-plastic elastomer ..." with interest. While the concept of healable dynamic networks as the tool to make polymers self heal has been known and applied for quite some time now (and the authors demonstrate to be fully aware of the on-going literature) this works demonstrates a new 'trigger' to induce/accelerate self healing. The experiments are well conducted and the assumed increased mobility of the ionic cluster is demonstrated convincingly using ESR.

The work itself is of limited industrial relevance and from a strictly polymer physical perspective it does not bring any new insight, but the concept of using a polar gas to control the mobility of the PDMS is sufficiently new and interesting to a larger audience to recommend publication

Reviewer #3 (Remarks to the Author):

In this work, authors have described the synthesis of an ionically cross-linked PDMS (ionomers) which exhibits gas-plastic behavior in the presence of CO₂ gas. In particular, authors demonstrate that the exposure to CO₂ gas results in the softening of the polymeric matrix and increased rate of ionic cross-linker exchange. This gas-selective modulation of mechanical properties is then utilized to demonstrate efficient self-healing when the damaged material is exposed to carbon dioxide.

I find the idea of using CO₂ as the healing agent interesting and intriguing. I point out that a few

other groups have used a thematically aligned but orthogonal approach to use carbon-fixation as a healing mechanism, e.g., <https://onlinelibrary.wiley.com/doi/10.1002/adma.201804037>. Additionally, CO₂ has been used in the design of self-healing cement and its plasticization effect on gas exchange membrane is also known in the literature (e.g., <https://pubs.acs.org/doi/full/10.1021/ie0204639>). I suggest that the authors expand the introduction and discuss these works and compare/contrast their abiotic approach with these methods when applicable.

The work is interesting but the following critical issues should be addressed to strengthen the work and expand its appeal to the broad audience of Nat. Commun:

- 1) The effect of water as a plasticizer is not discussed in the paper. How does the water impact mechanical properties and self-healing of these materials?

- 2) The exact mechanism of CO₂-mediated enhancement in the rate of exchange is not discussed/investigated in the paper. Authors have used a number of mechanical characterization techniques such as DMA, tensile, and rheology but these experiments only show that in the presence of CO₂ the material is softer/has a faster relaxation rate. At the molecular level, does CO₂ bind and react with carboxylate groups? How does this enhance the rate of cross-linker exchange? In particular, I find ESR experiment interesting (Figure 3a,b). However, there is only a modest difference in T_g between CO₂ and N₂. What is the local T_g in the air? Given that the T_g in all these cases is well below the experimental conditions (i.e., polymer already have significant mobility at RT), how is CO₂ increases the rate of cross-linker exchange?

- 3) To prove the generality of this concept, I suggest that authors modify the ionic group, from carboxylate to another group (i.e., sulfonated polymers or positively charged group) and demonstrate the broad applicability of gas-plastic concept.

Comments from Reviewer 1:

This manuscript reports on an elastomer that plasticized and self-healed by exposure to CO₂ gas. To my knowledge, this is the first report on a polymer that plasticized by a gas compound. Even though self-healing induced by pure gas compounds including CO₂ may not have high potential for practical use because such stimuli is not ubiquitous, processability at low temperature is appealing in terms of energy-saving process. More importantly, the “gas-plastic” behavior is scientifically valued and of high impact due to its novelty. I believe this manuscript can be appropriate for publication in Nature Communications after addressing the comments listed below:

1. The “Gas-plastic” property of the elastomer including self-healing behavior is clearly demonstrated with many experimental data. However, unfortunately, the authors pay less attention to the mechanism of the Gas-plastic behavior. Why is this elastomer softened by CO₂ gas? The schematic illustration in Fig. 1c only show the experimental results that CO₂ enhance the rearrangement rate of ionic crosslinks. The reason why this happens should be explained. Also, why does the rearrangement rate of ionic crosslinks change depending on the neutralization level?

Response: To reveal the gas-plastic behavior mechanism, we compared FT-IR spectra of the PDMS-80Na measured in air and CO₂ gas atmospheres. Fig. 3a shows the FT-IR result. The FT-IR spectra reveal that the intensity of the band indicating the stretching vibration of sodium carboxylate at 1580 cm⁻¹ decreases and the intensity of the COOH carbonyl stretching band at 1707 cm⁻¹ increases due to CO₂ exposure. This result clearly indicates that the polar CO₂ molecules permeate into the ionic aggregates and that the ionic bond between sodium cation and carboxylate anion is weakened by the polar CO₂ molecules, leading to the plasticization of the ionic aggregates. As a result of the weakened attractive force between the ionic sites, the network rearrangement rate increases as illustrated in Fig. 1c. This comment is added in the revised manuscript (lines 159–163 on page 10). Further, the network rearrangement rate increases with decreasing neutralization level because unneutralized carboxy groups act as plasticizers in the ionic aggregates. This result was originally demonstrated by Register et al. for polyethylene-based ionomers in ref. 41. This information is added on page 8 (lines 125–126) in the revised manuscript.

2. Why is PDMS-90Na so brittle compared with the other elastomers?

Response: When the neutralization level is very high, intramolecular aggregation of ionic groups is enhanced. In particular, this effect is remarkable for oligomeric PDMS used in this study. If the intramolecular ionic aggregation is enhanced, the elastomer would be brittle because the crosslinking chains between ionic aggregates decrease. A possible explanation to this idea has been added in the revised manuscript (lines 138–141 on page 9).

3. Healing test results are only given for PDMS-75Na and PDMS-80Na. Healing ability of the other samples should be also discussed.

Response: PDMS-*x*Na exhibits the self-healing ability irrespective of neutralization level. The self-healing behavior of the PDMS-60Na is exhibited in Fig. S13 in the revised Supplementary Information.

4. Exact pressure of the experimental conditions including healing conditions should be provided. "Normal pressure" (line 2 on page 6) is unclear.

Response: The pressure is approximately 0.1 MPa; this detail has been added page 4, line 63 of the revised manuscript.

5. The chemical structures in Fig. 1a, S1, and S2 seems to be incorrect. For example, the structural formula given in Fig. S1 indicates that every polymer chain of PDMS-NH₂ has only one aminopropylmethylsilane in the middle of the chain. However, I guess from the preparation method that PDMS-NH₂ is a simple random copolymer of dimethylsilane and aminopropylmethylsilane containing 2.7 mol% of the latter monomer on average.

Response: We corrected the chemical structure in Figs. 1a, S1, and S2 in the revised manuscript.

Comments from Reviewer 2:

I have read the manuscript "a gas-plastic elastomer ..." with interest.

While the concept of healable dynamic networks as the tool to make polymers self heal has been known and applied for quite some time now (and the authors demonstrate to be fully aware of the on-going literature) this work demonstrates a new 'trigger' to induce/accelerate self healing. The experiments are well conducted and the assumed increased mobility of the ionic cluster is demonstrated convincingly using ESR.

The work itself is of limited industrial relevance and from a strictly polymer physical perspective it does not bring any new insight, but the concept of using a polar gas to control the mobility of the PDMS is sufficiently new and interesting to a larger audience to recommend publication

Comments from Reviewer 3:

In this work, authors have described the synthesis of an ionically cross-linked PDMS (ionomers) which exhibits gas-plastic behavior in the presence of CO₂ gas. In particular, authors demonstrate that the exposure to CO₂ gas results in the softening of the polymeric matrix and increased rate of ionic cross-linker exchange. This gas-selective modulation of mechanical properties is then utilized to demonstrate efficient self-healing when the damaged material is exposed to carbon dioxide.

I find the idea of using CO₂ as the healing agent interesting and intriguing. I point out that a few other groups have used a thematically aligned but orthogonal approach to use carbon-fixation as a healing mechanism, e.g., <https://onlinelibrary.wiley.com/doi/10.1002/adma.201804037>. Additionally, CO₂ has been used in the design of self-healing cement and its plasticization effect on gas exchange membrane is also known in the literature (e.g.,

<https://pubs.acs.org/doi/full/10.1021/ie0204639>). I suggest that the authors expand the introduction and discuss these works and compare/contrast their abiotic approach with these methods when applicable.

The work is interesting but the following critical issues should be addressed to strengthen the work and expand its appeal to the broad audience of Nat. Commun:

Response: References 36, 37, 42, 43, 44 are additionally cited in text, as per your suggestions.

1) The effect of water as a plasticizer is not discussed in the paper. How does the water impact mechanical properties and self-healing of these materials?

Response: We performed additional experiments on the fully moisture-absorbed PDMS-80Na. Because of the hydrophilic property of the ionic aggregates, the moisture plasticizes the ionic aggregates and increases the network rearrangement rate as shown in Fig. S12d. Thus, the moisture reduces the mechanical strength of the elastomer, as shown in Fig. S10. Moreover, the moisture is expected to accelerate the self-healing because of the increased network rearrangement rate. These phenomena are explained in the revised manuscript (lines 157–158 on page 10).

2) The exact mechanism of CO₂-mediated enhancement in the rate of exchange is not discussed/investigated in the paper. Authors have used a number of mechanical characterization techniques such as DMA, tensile, and rheology but these experiments only show that in the presence of CO₂ the material is softer/has a faster relaxation rate.

At the molecular level, does CO₂ bind and react with carboxylate groups? How does this enhance the rate of cross-linker exchange?

In particular, I find ESR experiment interesting (Figure 3a,b). However, there is only a modest difference in T_g between CO₂ and N₂. What is the local T_g in the air?

Given that the T_g in all these cases is well below the experimental conditions (i.e., polymer already have significant mobility at RT), how is CO₂ increases the rate of cross-linker exchange?

Response: To reveal the gas-plastic behavior mechanism, we compared FT-IR spectra of the PDMS-80Na measured in air and CO₂ gas atmospheres. Fig. 3a shows the FT-IR result. The FT-IR spectra reveal that the intensity of the band indicating the stretching vibration of sodium carboxylate at 1580 cm⁻¹ decreases and the intensity of the COOH carbonyl stretching band at 1707 cm⁻¹ increases due to CO₂ exposure. This result clearly indicates that the polar CO₂ molecules permeate into the ionic aggregates and that the ionic bond between sodium cation and carboxylate anion is weakened by the polar CO₂ molecules, leading to the plasticization of the ionic aggregates. As a result of the weakened attractive force between the ionic sites, the network rearrangement rate increases, as illustrated in Fig. 1c. This information is added in the revised manuscript (lines 159–163 on page 10). We also measured the ESR in the air. The result is shown in Fig. 3b.

3) To prove the generality of this concept, I suggest that authors modify the ionic group, from carboxylate to another group (i.e., sulfonated polymers or positively charged group) and demonstrate the broad applicability of gas-plastic concept.

Response: The reviewer's suggestion favors with our interest. We are currently conducting experiments, and the results will be reported in near future.

REVIEWERS' COMMENTS:

Reviewer #1 (Remarks to the Author):

The revised manuscript is now worth publishing in Nat. Commun.

Reviewer #3 (Remarks to the Author):

Authors have answered most of my questions in their revisions. However, there are still two issues that should be addressed:

1) CO₂ is not a polar gas. Strictly speaking, the dipoles are canceled due to the symmetry of the molecules. Maybe a different rationalization/or term should be used throughout the paper to discuss the effect of the CO₂ plasticization. I suggest dielectric constant or some other related terms.

2) I appreciate that the authors have performed additional experiments to elucidate the mechanism of the gas-plasticity. I have read their response to my questions and reviewer #1. However, I can not fully follow their interpretation of the experimental data (FT-IR figure 3a). In particular, this reviewer disagrees that "this result ****clearly**** indicates that the polar CO₂ molecules permeate into the ionic aggregates and that the ionic bond between sodium cation and carboxylate anion is weakened by the polar CO₂ molecules, leading to the plasticization of the ionic aggregates"

Since this question is brought up by two reviewers, I think addressing it more clearly in the manuscript can be beneficial to readers as well. I am not asking for additional experiments and authors can use the current data to further explain this phenomenon.

According to the reviewer's comments, we revised our manuscript, and our answers to the reviewer's questions are as follows:

Comments from Reviewer 3:

Authors have answered most of my questions in their revisions. However, there are still two issues that should be addressed:

1) CO₂ is not a polar gas. Strictly speaking, the dipoles are canceled due to the symmetry of the molecules. Maybe a different rationalization/or term should be used throughout the paper to discuss the effect of the CO₂ plasticization. I suggest dielectric constant or some other related terms.

Response: I apologize for the misses. I used the term "polarizability" in the revised manuscript (line 161 on page 11).

2) I appreciate that the authors have performed additional experiments to elucidate the mechanism of the gas-plasticity. I have read their response to my questions and reviewer #1. However, I can not fully follow their interpretation of the experimental data (FT-IR figure 3a). In particular, this reviewer disagrees that "this result ****clearly**** indicates that the polar CO₂ molecules permeate into the ionic aggregates and that the ionic bond between sodium cation and carboxylate anion is weakened by the polar CO₂ molecules, leading to the plasticization of the ionic aggregates"

Since this question is brought up by two reviewers, I think addressing it more clearly in the manuscript can be beneficial to readers as well. I am not asking for additional experiments and authors can use the current data to further explain this phenomenon.

Response: From the change in the FT-IR bands, which are assigned to the stretching vibrations of sodium carboxylate and carboxy groups by the CO₂ exposure (Fig. 3a), the penetration of CO₂ molecules into the ionic aggregates is a credible claim. However, the precise interaction mechanism between the CO₂ molecule and sodium carboxylate is hard to discuss from only the FT-IR spectrum. Therefore, we revised our manuscript as follows: "The FT-IR bands at 1586 cm⁻¹ and 1716 cm⁻¹ assigned to the stretching vibrations of sodium carboxylate and carboxy groups, respectively, are affected by CO₂ exposure (Fig. 3a). This demonstrates that the CO₂ molecules with relatively high polarizability penetrate into the ionic aggregates, leading to the plasticization of the ionic aggregates." (lines 158–162 on page 10)